chemical physics

silicene, electronic properties, *p*-type doping, first-principles, substitution

**Author for correspondence:**
Ming-Fa Lin
e-mail: mflin@mail.ncku.edu.tw

# Rich *p*-type-doping phenomena in boron-substituted silicene systems

Hai Duong Pham[1], Wu-Pei Su[2], Thi Dieu Hien Nguyen[3], Ngoc Thanh Thuy Tran[4] and Ming-Fa Lin[3]

[1]Center of General Studies, National Kaohsiung University of Science and Technology, Kaohsiung, Taiwan
[2]Department of Physics, University of Houston, Houston, 77204 TX, USA
[3]Department of Physics, and [4]Hierarchical Green-Energy Materials (Hi-GEM) Research Center, National Cheng Kung University, Tainan 70101, Taiwan

(iD) M-FL, 0000-0002-6255-1444

The essential properties of monolayer silicene greatly enriched by boron substitutions are thoroughly explored through first-principles calculations. Delicate analyses are conducted on the highly non-uniform Moire superlattices, atom-dominated band structures, charge density distributions and atom- and orbital-decomposed van Hove singularities. The hybridized $2p_z$–$3p_z$ and [2s, $2p_x$, $2p_y$]–[3s, $3p_x$, $3p_y$] bondings, with orthogonal relations, are obtained from the developed theoretical framework. The red-shifted Fermi level and the modified Dirac cones/$\pi$ bands/$\sigma$ bands are clearly identified under various concentrations and configurations of boron-guest atoms. Our results demonstrate that the charge transfer leads to the non-uniform chemical environment that creates diverse electronic properties.

## 1. Introduction

It is well known that the theoretical predictions could be achieved with numerical simulations and the phenomenological models. For example, the former and the latter, respectively, cover the first-principles method, molecular dynamics or Monte Carlo [1–3] and the tight-binding model, the effective-mass approximation, random-phase approximation [4–6]. Specifically, the Vienna ab initio simulation package (VASP) calculations are capable of dealing with condensed-matter systems with many atoms in a unit cell [7], multi-orbital chemical bondings, spin-dependent interactions [8] and on-site Coulomb interactions and composite structures [9]. With numerical methods, a theoretical framework will be developed to comprehend the critical physical/chemical/ material mechanisms. On the other hand, the generalized tight-binding model [10], being very suitable under a uniform

perpendicular magnetic field, is successfully established for layered graphene systems [11]. The diverse phenomena of magnetic quantization are clearly revealed in the various two-dimensional emergent materials [12], such as the unusual magnetic properties of group-IV [13] and group-V [14] few-layer systems.

As for two-dimensional emergent materials, the layered group-IV [15] and group-V [16] systems have been successfully synthesized through different growth techniques. Very interestingly, such condensed-matter systems are outstanding candidates for basic and applied sciences [17], integrated engineering [18] and highly efficient products [19], mainly owing to the diversified physical/chemical/material phenomena [20,21]. The emergence of recent synthetic experiments has shown that few-layer graphene systems can be produced by various methods, e.g. mechanical exfoliation [22] and molecular vapour deposition [23]. However, their partners, two-dimensional silicene, germanene, tinene or Pb systems can be generated only with the molecular beam epitaxy (MBE) [24]. According to previous studies, the essential properties are easily modulated by important factors, i.e. they are quite sensitive to changes of the lattice symmetries [25], layer [26], sequence stacking configurations [27], gate voltages [28], magnetic fields [29], mechanical strains [30] and chemical reactions [31].

Nearly all two-dimensional materials possess very active surface/edge structures, so chemical modifications could be used to create dramatic transformations of fundamental properties, especially the semiconductor–semimetal/metal transitions [32], and those among the non-magnetic, ferromagnetic and anti-ferromagnetic spin configurations [33]. The graphene-related compounds with adatom/molecule chemisorptions [34] or guest-atom substitutions [35], have been clearly identified to exhibit the diverse phenomena [36] and have many potential applications [37]. The hydrogenated [38], halogenated [39], alkalized [40], aluminized [41] or oxidized [42] graphene systems are examples of those compounds. Recent researches on silicene-based materials cover some isolated experimental [43] and theoretical works [44]. For example, the rare-earth adlayers on silicene surface exhibit unusual transition of the anti-ferromagnetic and ferromagnetic configurations during a decrease of the adlayer thickness [45], and a two-dimensional magnet is realized from buckled silicene. Moreover, boron-substituted silicene, as done under a specific guest-atom distribution, is predicted to become a $p$-type metal [46,47]. However, the mere mention of a single concentration or configuration in previous studies makes the dependence of these features unclear. Apparently, the critical conclusions and mechanisms could be only achieved through systematic investigations.

The rich and unique phenomena clearly revealed in the two-dimensional binary boron-substituted silicene compounds [48] are thoroughly explored with the delicate calculations and analyses. Furthermore, the theoretical framework is further developed to demonstrate possible extensions. The first-principles method is employed to understand the geometric, electronic and magnetic properties, and the sensitive dependences on the concentrations and configurations of boron-host atoms are investigated in detail. The calculated results can provide sufficient information to identify the critical mechanisms. They encompass the ground state energies, the various Si–Si, Si–B and B–B chemical bond lengths, the Si- and B-co-dominated valence and conduction bands, the spatial charge densities, and the atom- and orbital-decomposed van Hove singularities. In addition, the spin-dependent interactions are included in the numerical evaluations to see whether the magnetic configurations play an important role in the essential properties. Most importantly, the concise pictures of multi-orbital hybridizations, which are related to the existence of modified Dirac-cone structure, the $p$- or $n$-type dopings and free carrier density, and the regular or irregular $\pi$ and $\sigma$ bondings, are proposed accurately. Moreover, the relationship between VASP and the tight-binding model [49] in band structures is discussed.

## 2. Method of calculation

The properties of boron-substituted silicene are studied by density functional theory (DFT) implemented in Vienna ab initio simulation package (VASP) [50,51]. The exchange and correlation energies, which come from the many-particle Coulomb interactions, were calculated with the use of the Perdew–Burke–Ernzerhof (PBE) functional under the generalized gradient approximation [52]. Furthermore, the projector augmented wave (PAW) pseudopotentials can characterize the electron–ion interactions [53]. The cut-off energies of the wave function expanded by plane waves were chosen to be 500 eV, whereas to avoid the interaction between adjacent cells, vacuum distance along $z$-axis is set to be 15 Å. The first Brillouin zone (BZ) is sampled in the Monkhorst–Pack scheme along the two-dimensional periodic direction by $9 \times 9 \times 1$ k-points for structure relaxations. For accurate calculation of electronic properties, we used $100 \times 100 \times 1$ k-points for high concentration cases (50 and 100%) and $30 \times 30 \times 1$ for lower concentration configurations (the rest of the cases). During the ionic relaxations process, the unit vectors

Si-atom                                    B-atom

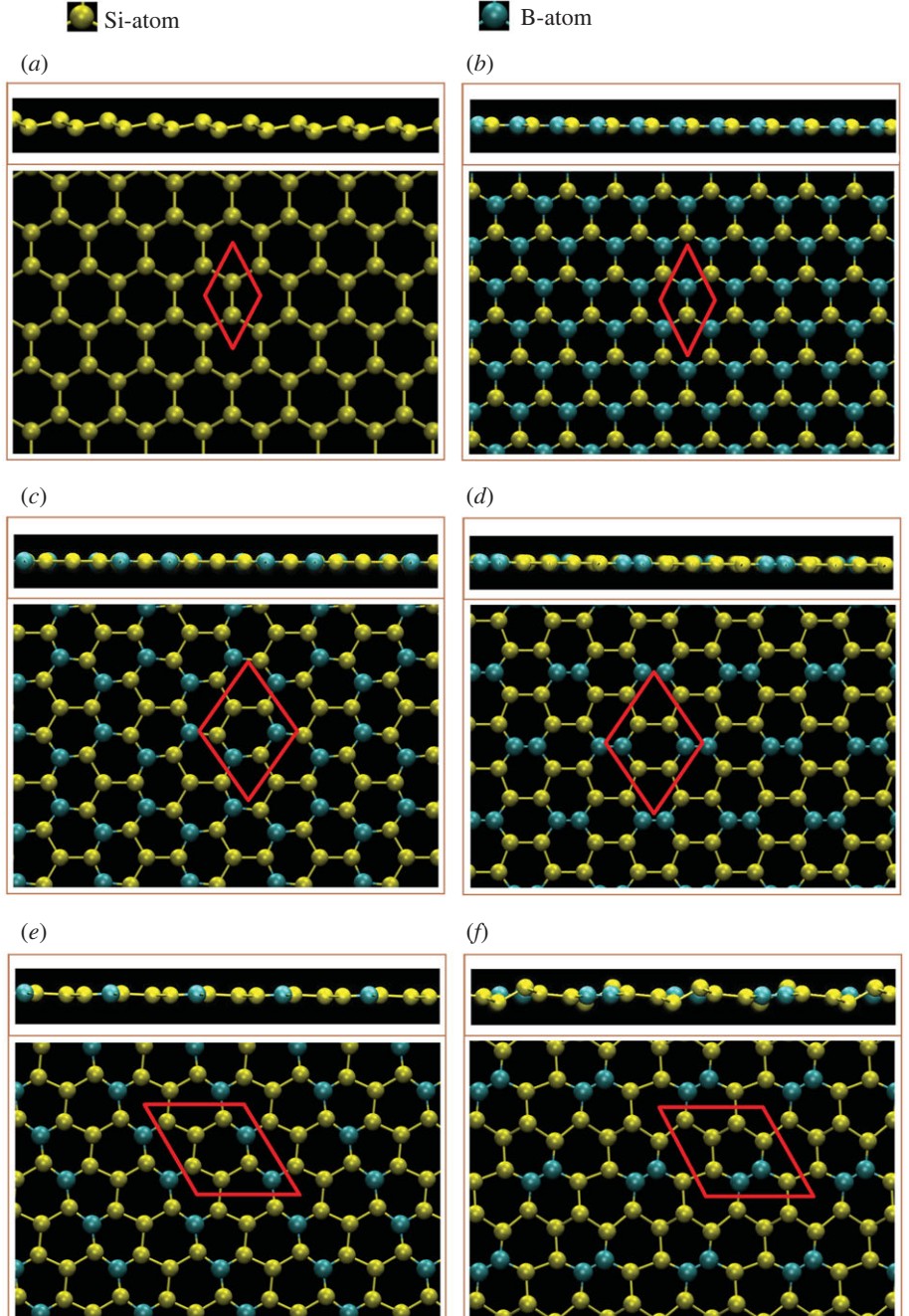

**Figure 1.** The side and top views of optimal geometric structures for boron-substituted silicene systems with (*a*) pristine, (*b*) 100%-, (*c*) 50%-meta-, (*d*) 50%-ortho-/para-, (*e*) 33.3%-meta, (*f*) 33%-ortho, (*g*) 33%-para- (*h*) 12.5%-meta-, (*i*) 12.5%-ortho-, (*j*) 12.5%-para. (*k*) 14.3%-single- and (*l*) 5.7%-single-substitutions, the red parallelogram/rhombus represent the unit cell.

are kept fixed, the maximum Hellmann–Feynman force acting on each atom is less than 0.01 eV Å$^{-1}$ and the convergent energy scale was chosen to be $10^{-5}$ eV between two consecutive steps.

# 3. Results and discussion

## 3.1. Geometric structures and electronic band structures

For the boron-substituted silicene systems thus studied (figure 1*a–l*), all the intrinsic interactions are included in the delicate first-principles calculations. Four typical configurations including meta, ortho, para and single cases are chosen for a model investigation. The lowest and highest ground state

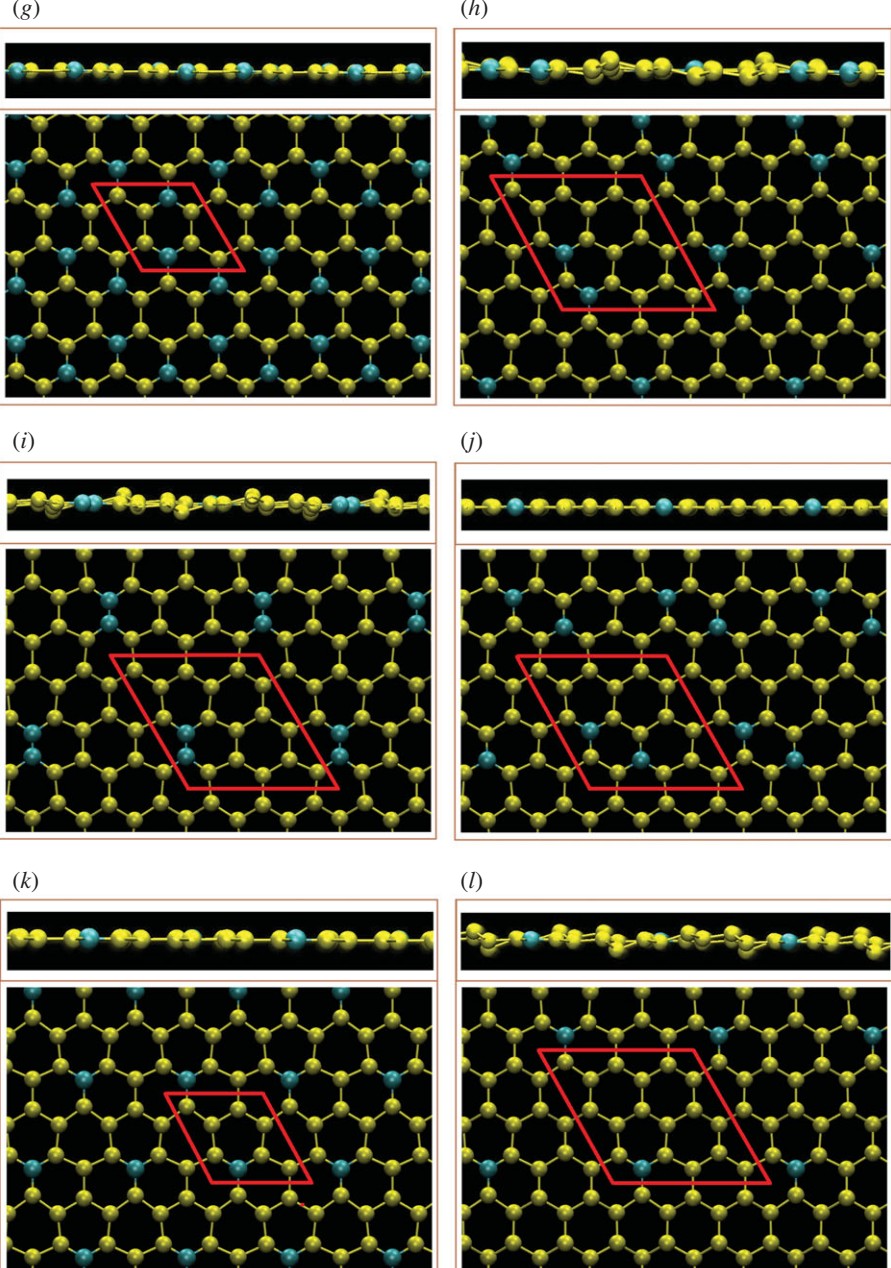

**Figure 1.** (*Continued.*)

energies are demonstrated by the third and fourth types, respectively. The most stable one belongs to the para configuration and this aspect can be considered when the systems are synthesized in experimental growths. However, the B- and Si-dependent spin configurations/interactions are fully suppressed by the very strong and complicated chemical bondings. That is to say, the spin-induced magnetic configuration is absent in any B-substituted silicene. Compared with the pristine silicene, an enlarged unit cell is mainly determined by the B-substitution concentration and configuration. The uniform chemical bonding in a Moire superlattice could survive only under the [1:1] condition with the three nearest-neighbouring B–Si bonds embedded in a planar honeycomb lattice (table 1), with a bond length of approximately 1.95 Å. With decreasing B-concentrations, other modulated chemical environments are possible including B–Si and Si–Si bonds for all cases, as well as B–B ones only for ortho-cases. Apparently, the B–Si bond length is roughly the average of B–B and Si–Si bond lengths. This could be attributed to the well-separated/-defined $\pi$ and $\sigma$ bondings. That indicates the negligible contributions of the $sp^3$ bondings. More clearly, the weak/absence of $sp^3$ bondings can be realized by most of the high B-concentrations exhibiting the planar geometric structure. When the boron-concentration decreases,

**Table 1.** The optimal geometric structures of boron-substituted silicene systems under various concentrations and configurations in terms of the B–B, B–Si and Si–Si bond lengths, the band gaps, ground state energies per unit cell, and buckling between A and B sublattices.

| number atoms/ unit cell | percentage | ratio of B–Si | B–B bond length (Å) | B–Si bond length (Å) | nearest Si–Si (Å) | second nearest Si–Si (Å) | $E_g^d$(eV) /metal | optimization $E_0$ (eV) | buckling $\Delta$ (Å) |
|---|---|---|---|---|---|---|---|---|---|
| 2 | X | Pristine | X | X | 2.250 | X | 0.001 | X | 0.48 |
| 6 | 100 | 1:1 | X | 1.953 | X | X | metal | − 10.427 | 0 |
| | 50 | 2:4 meta | X | 1.963 | 2.290 | X | metal | − 30.640 | 0 |
| | 50 | 2:4 ortho/para | 1.639 | 1.965 | 2.295 | X | metal | − 30.560 | 0 |
| | 20 | 1:5 | X | 1.923 | 2.262 | X | metal | − 29.831 | 0 |
| 8 | 33.3 | 2:6 meta | X | 1.943–1.992 | 2.239 | 2.280 | metal | − 40.022 | 0.07 |
| | 33.3 | 2:6 ortho | 1.717 | 1.997 | 2.294 | 2.256 | metal | − 39.643 | 0.34 |
| | 33.3 | 2:6 para | X | 1.945 | 2.281 | X | metal | − 40.356 | 0 |
| | 14.3 | 1:7 | X | 1.969 | 2.268 | 2.233 | metal | − 39.217 | 0 |
| 18 | 12.5 | 2:16 meta | X | 1.969–2.002 | 2.279 | 2.259–2.268 | metal | − 87.851 | 0.24–0.64 |
| | 12.5 | 2:16 ortho | 1.712 | 2.011 | 2.284–2.213 | 2.269–2.307 | metal | − 87.463 | 0.28–0.6 |
| | 12.5 | 2:16 para | X | 1.947 | 2.266–2.292 | 2.236–2.256 | metal | − 88.161 | 0.07–0.15 |
| | 5.8 | 1:17 | X | 1.961 | 2.283 | 2.268–2.280 | metal | − 86.982 | 0.2–0.6 |

the buckling will gradually recover. Most importantly, two and three kinds of chemical bonds lead to non-uniform chemical environment in an enlarged unit cell, as clearly illustrated by the fact that the B–Si and Si–Si bond lengths and even the height differences of A and B sublattice show significant fluctuations (table 1). This behaviour is expected to create complicated hopping integrals/orbital hybridizations in the phenomenological models [54].

The boron-substituted silicene systems present rich and unique band structures, mainly owing to the three valence electrons of B-atom in the [2s, $2p_x$, $2p_y$, $2p_z$] orbitals. In figure 2, all the electronic states are plotted along $\Gamma$-K-M-$\Gamma$ paths in the Brillouin zone scheme for many unit cells. Pristine silicene (figure 2a) presents a narrow bandgap of $E_g = 0.001$ eV mainly due to the spin–orbital coupling. This result is quite consistent with that obtained in previous studies [55–57]. The full-substitution [1:1] two-dimensional boron–silicon compound, with a planar honeycomb lattice, is an unusual two-dimensional metal/semimetal, as clearly shown in figure 2b. The boron and silicon atoms (the red triangles and blue circles) have comparable weights for all valence and conduction bands. The asymmetric energy spectrum of valence and conduction bands about the Fermi level is greatly enhanced by boron substitution, since $E_F$ intersects with the former. Plenty of free valence holes are created near the stable K and $\Gamma$ valleys simultaneously. The $\pi/\pi^*$ and the first $\sigma$ bands could be well defined, according to the modified Dirac-cone structure and the concave-downward parabolic dispersions, respectively. The $\pi$- and $\pi^*$-electronic energy spectra emanate from the K point, respectively, at $E^v = 1.70$ eV and $E^c = 2.00$ eV with a gap of approximately 0.30 eV. Compared with that of a pristine silicene (figure 2a), the Dirac-cone band structure is strongly modulated by the impure hybridizations of $2p_z$–$3p_z$ orbitals. Obviously, the $\pi$-band energy width grows from 3.20 to 4.00 eV. On the other hand, the hybridized $\sigma$ bondings of [2s, $2p_x$, $2p_y$]–[3s, $3p_x$, $3p_y$] orbitals could affect the dispersions, e.g. the energy and number of band-edge states. The doubly degenerate $\sigma$-band state at the $\Gamma$ point is located slightly above the Fermi level, so that the unoccupied electronic states close to it become free valence holes. The first and second $\sigma$-bands, which are continuous along the $\Gamma$MK direction, have a width of approximately 4.30 eV and 6.20 eV, respectively. The anti-crossing/crossing behaviours of the $\sigma$ and $\pi$ bands further indicated that their chemical bondings are orthogonal to each other in a planar honeycomb B–Si lattice. That is to say, the anti-crossing-induced van Hove singularities are absent in the orbital and atom-projected density of states (discussed later in figure 4a). Most importantly, the $\pi$ and $\sigma$ energy bands are well characterized in the full boron substitution, as revealed in monolayer silicene [58,59] graphene.

Obviously, the concentrations and configurations of boron guest atoms play a critical role in diversifying the electronic properties, as shown in figure 2a–l. They are able to dominate/specify the rich multi-orbital hybridizations of the various chemical bondings and thus create the unusual band structures. The main features of band structures are very sensitive to the B-substitution, in which the strongly modified properties include the asymmetry of hole and electron energy spectra about the Fermi level, the non-uniform dominances of boron and silicon atoms, the initial valence and conduction states from the K and $\Gamma$ valleys, the size of the energy gap in the modified Dirac cone structure, the $\pi$- and $\sigma$-band widths (the initial and final state energies), the number of band-edge states and the two-dimensional free hole densities. Very interestingly, an almost gapless/perfect Dirac-cone structure in a pristine silicene (figure 2a) becomes seriously deformed under the various B-substitution cases. The Dirac-cone energy gap might depend on the different ionization energies (the distinct site energies) of the B-$2p_z$ and Si-$3p_z$ orbitals [60]. Whether the modified Dirac-cone structures come to exist at the K or $\Gamma$ point is related to the zone-folding effect (the enlarged unit cell) [61]. That is to say, the original hexagonal first Brillouin zone is folded into the smaller one according to the ratio of the enlarged superlattice volume versus the pristine one. As a result, the pristine corner point might correspond to the same one or the origin in the reduced first-BZ. The $\pi/\pi^*$ valence/ conduction energy sub-band presents a saddle-point structure near the M point with a prominent density of states. The valence and conduction Dirac cones are, respectively, dominated by boron and silicon atoms for the [1:1] case (figure 2b). The boron dominance disappears for any other B-concentration (figure 2c–l) (also the density of states in figure 4b–l). With decreasing boron-guest atom concentrations, there are more valence and conduction sub-bands ($\pi$- and $\sigma$-electronic ones), leading to the frequent sub-band crossings. However, few anti-crossings of the $\sigma$ valence bands could survive at $E^v < -2.3$ eV. Such phenomena, which might rely on the hybridized $\sigma$ bondings of [2s, $2p_x$, $2p_y$] – [3s, $3p_x$, $3p_y$] orbitals, could create the constant-energy loops (the quasi-one-dimensional parabolic dispersions) in the energy–wave–vector space. Of course, the boron-concentration is lower, and so the two-dimensional free hole density, as observed in the reduced first Brillouin zone. In short, the unusual non-uniform chemical/physical environments in a Moire superlattice, which mainly originate from the impure/pure $\pi$ and $\sigma$-electronic orbital hybridizations in the distinct chemical

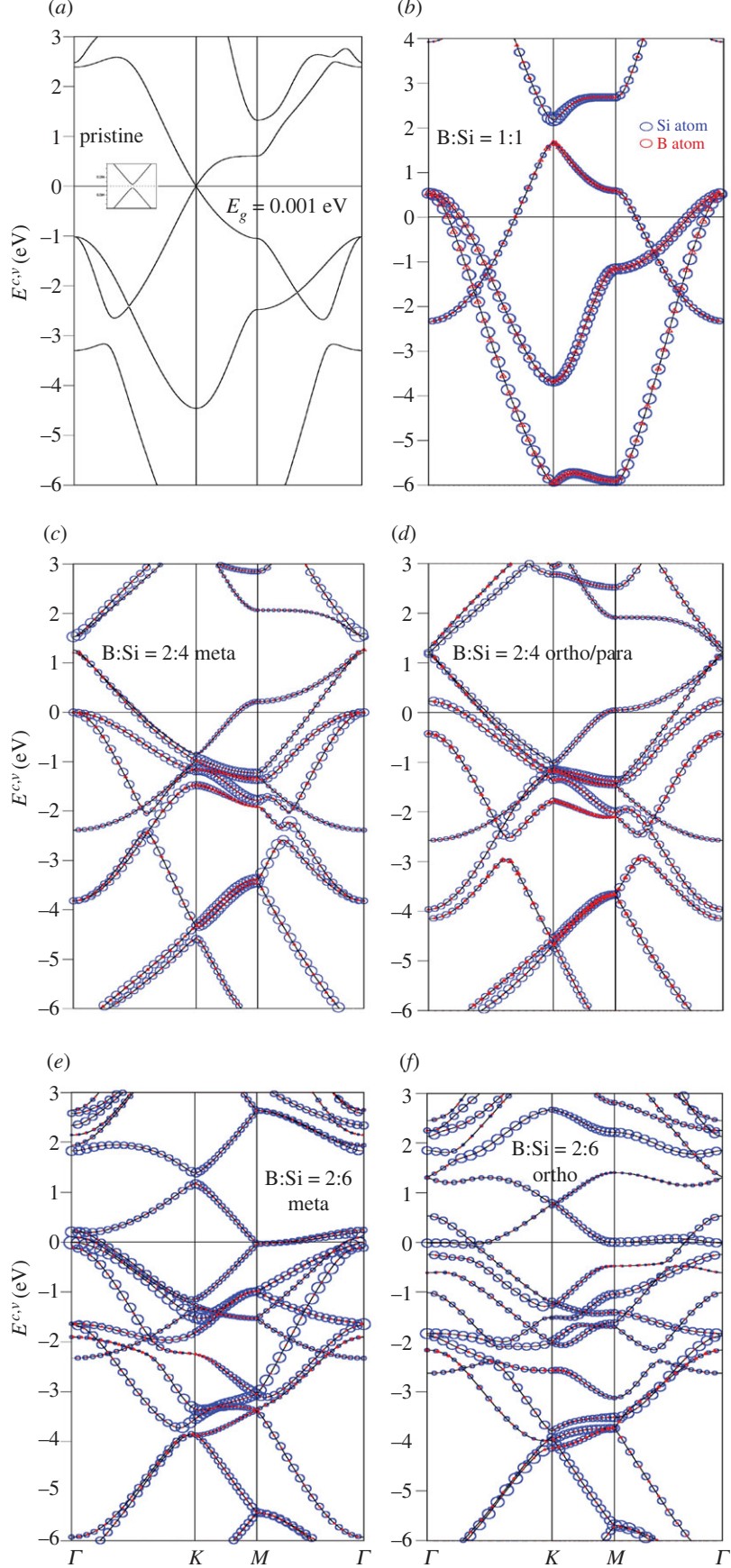

**Figure 2.** Band structures of the B-substituted silicene materials, with the Si- and B-dominances (the blue and red circles) under the various cases: (a) without substitution, (b) [1:1]-, (c) [2:4]-meta-, (d) [2:4]-ortho-/para-, (e) [2:6]-meta-, (f) [2;6]-ortho-, (g) [2:6]-para- (h) [2:16]-meta-, (i) [2:16]-ortho-, (j) [2:16]-para-, (k) [1:7]-single-, (l) [1:17]-single-substitution configurations.

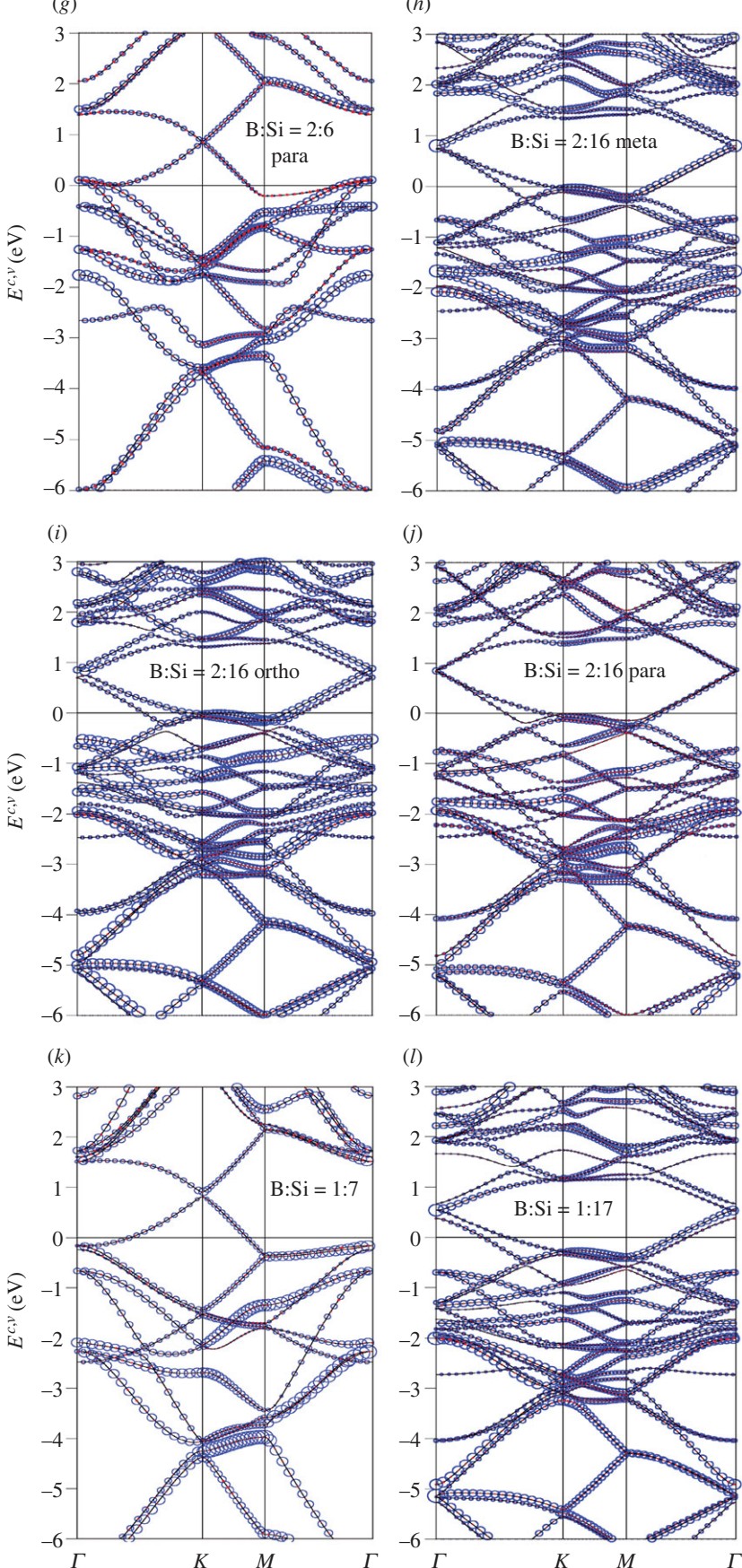

**Figure 2.** (*Continued.*)

bonds, are responsible for the diversified electronic energy spectra. The high-resolution angle-resolved photoemission spectroscopy (ARPES) measurements on the main features of occupied states are very useful in examining the B-guest-atom substitution effects, such as the concentration- and configuration-dependent $\pi$ and $\sigma$ valence bands, the modified Dirac-cone structures with anisotropic dispersions and Fermi momenta, and the unoccupied hole states near the stable $K$ or $\Gamma$ valleys.

In this study, all the B-substituted silicene systems are predicted to exhibit the $p$-type doping phenomena, in which the two-dimensional free hole density is deduced to be the same as that of guest B-atoms. Boron and silicon atoms, respectively, possess three and four valence electrons, while they create the complex $\pi$- and $\sigma$-chemical bondings through the $2p_z$–$3p_z$/$2p_z$–$2p_z$/$3p_z$–$3p_z$ and [2s, $2p_x$, $2p_y$]–[3s, $3p_x$, $3p_y$]/[2s, $2p_x$, $2p_y$]–[2s, $2p_x$, $2p_y$]/[3s, $3p_x$, $3p_y$]–[3s, $3p_x$, $3p_y$] orbital hybridizations. The well-separated $\pi$ and $\sigma$ states have been confirmed from the calculated band structures (figure 2$a$–$l$). Generally speaking, the pure or mixed $\pi$- and $\sigma$-bondings require four valence orbitals in each atom. This will lead to the unusual $p$-type doping because of only three valence orbitals in boron. According to the conservation of electronic states, the **k**-space area of the unoccupied valence states is equal to the boron-atom number per unit cell. For example, compared with a pristine system (figure 2$a$), the [1:1] B-substituted silicene (figure 2$b$) presents the same number of valence and conduction bands, but the Fermi levels are different: 0 and 1.80 eV for the former and the latter. Such result should be independent of the distribution and concentration of boron-host atoms. Based on the lower-energy valence bands in figure 2$b$–$l$, most of free holes are provided by the $\pi$ band near the $K/\Gamma$ valley (the hybridized/pure $\pi$ bondings), and part of them originate from the $\sigma$ bonds close to the $\Gamma$ valley (the $\sigma$ bondings due to the hybridizations of $2p_x$, $2p_y$, $3p_x$ and $3p_y$ orbitals), as further confirmed from the atom- and orbital-projected density of state in figure 4. Most importantly, they clearly indicate the greatly reduced $\pi$ bondings in the boron-substituted environments.

## 3.2. Spatial distribution of charge density

Certain important features of the electronic energy spectra in boron-substituted silicene systems could be understood from their charge density distributions, as clearly displayed in figure 3$a$–$h$ with the [$x$, $z$]- and [$x$, $y$]-plane projections. For the full substitution, the only B–Si bonds, which create a planar honeycomb lattice, present the unusual carrier density (figure 2$b$). The $\sigma$ bonding between the guest and host atoms, the [2s, $2p_x$, $2p_y$]–[3s, $3p_x$, $3p_y$] orbital hybridization, is obviously revealed in the red region. Furthermore, it is an asymmetric distribution relative to the bond centre. The higher carrier density is localized around the B-guest atoms, indicating the charge transfer from silicon to boron atoms. The $\sigma$-carrier density is accompanied by $\pi$ bonding (the $2p_z$–$3p_z$ orbital hybridization) at the outer region under the almost orthogonal condition. Roughly speaking, the spatial charge distribution of the full B-substitution is highly anisotropic even only in the presence of B–Si bonds, where its density decays quickly in going from the hexagonal sides to the centre of honeycomb lattice. The above-mentioned features have led to the changes in the $\sigma$- and $\pi$-electronic energy spectra (figure 2$b$), such as the modified Dirac-cone structure and the direct crossings of the $\pi$ and $\sigma$ valence sub-bands. However, the red shift of the Fermi level, corresponding to the valence Dirac cone (the creation of free holes in valence ones) are very difficult to directly identify from the current charge density distribution.

Most importantly, the $\pi$ and $\sigma$ bondings seem to be well characterized under the various boron-guest atom concentrations and configurations, as clearly illustrated in figure 3$c$–$h$. For the non-full boron substitutions, there exist B–Si, Si–Si and B–B bonds, in which they show the obvious $\sigma$ chemical bondings in the [$x$, $z$] and [$x$, $y$] planes. That is, the pure and impure $\sigma$-electronic hybridizations, which induce high density distribution between two atoms, come to exist in different bonds. Similar phenomena could also be found in the $\pi$ bondings. As a result, $\pi$ and $\sigma$ chemical bondings are well defined for all the boron-substitution cases. This phenomenon indicates a the tight-binding model [62] might be reliable in simulating the first-principles band structures (figure 2$a$–$l$) in which the atom- and orbital-dependent hopping integrals and site energies directions could be obtained from the well-fitted energy spectra along the high-symmetry points. The consistency of two methods is very useful in the further studies on the other essential properties, e.g. the optical [63], magnetic [64] and transport properties [65] of guest-atom-substituted silicenes/germaneness.

## 3.3. Density of states

The atom- and orbital-decomposed density of states, as clearly illustrated in figure 4$a$–$l$, are capable of providing more information about the $p$-type doping phenomena and the $\pi$-/$\sigma$-band widths of the

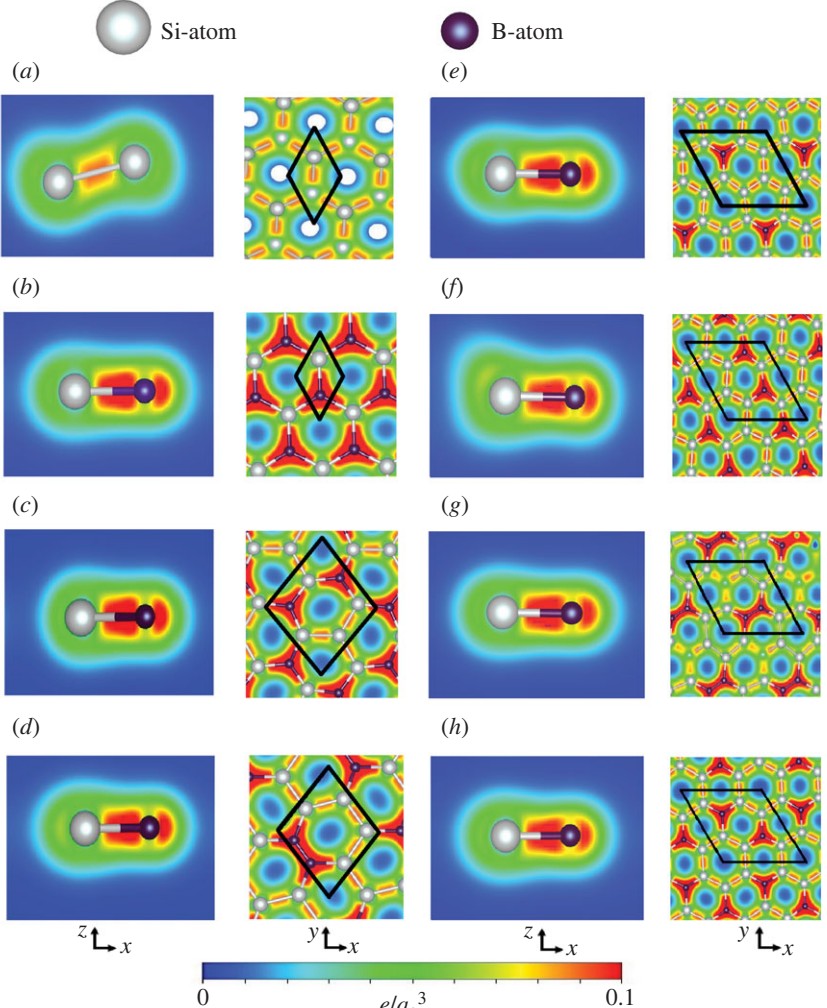

**Figure 3.** The spatial charge distributions in the boron-substituted silicene systems under various concentrations and configurations: (*a*) without boron-guest atoms, (*b*) [1:1], (*c*) [2:4]-meta, (*d*) [2:4]-ortho-/para, (*e*) [2:6]-meta, (*f*) [2;6]-ortho, (*g*) [2:6]-para and (*h*) [1:7] cases, with the [*x*, *z*]- and [*x*, *y*]-plane projections of the well-defined $\pi$ and $\sigma$ chemical bondings, the unit cells are represented by the black parallelogram/rhombus.

boron-substituted silicene materials. The two-dimensional energy spectra can create five kinds of van Hove singularities, the distorted V-shapes with/without observable energy gaps, shoulders, logarithmic symmetric peaks, square-root-form asymmetric peaks and delta-function-like peaks, respectively, corresponding to the modified Dirac cones, extremal points (local minima or maxima), saddle points, constant-energy loops (quasi-one-dimensional parabolic dispersions) and partially flat bands. Apparently, the low-lying density of states, being characterized by the modified Dirac-cone structure across the Fermi level, is mainly determined by the Si-$3p_z$ and B-$2p_z$ orbitals with merged van Hove singularities under any substitution cases (the solid purple curves and the dashed wine curves). That clearly indicates the dominances of the pure/hybridized $\pi$ bondings on the low-energy essential properties. The B-$3p_z$ orbitals make larger contribution to the valence Dirac-cone state only in the specific [1:1] case, while the opposite is true for the other cases. Furthermore, the Si-$3p_z$ orbitals dominate the conduction-cone energy spectrum for all cases. The pure and hybridized $\pi$ bondings create a pair of sharp shoulders with an observable energy spacing or an asymmetric V-shape structure with distinct slopes mainly owing to the modified Dirac cone (figure 2*b–l*). In general, the energy and spacing of the valence and conduction Dirac points decline quickly with decreasing B concentration.

The left-hand special structure corresponds to the initial $\pi$-electronic valence states at the *K* or $\Gamma$ valley. At deeper energies along the *KMΓ* or *ΓMK* path, prominent symmetric peaks/peak come to exist in the logarithmic form. Such structures are due to the split saddle-point valence states; furthermore, their positions might be above, near or below the Fermi level, depending on the critical case of [2:6]. As for

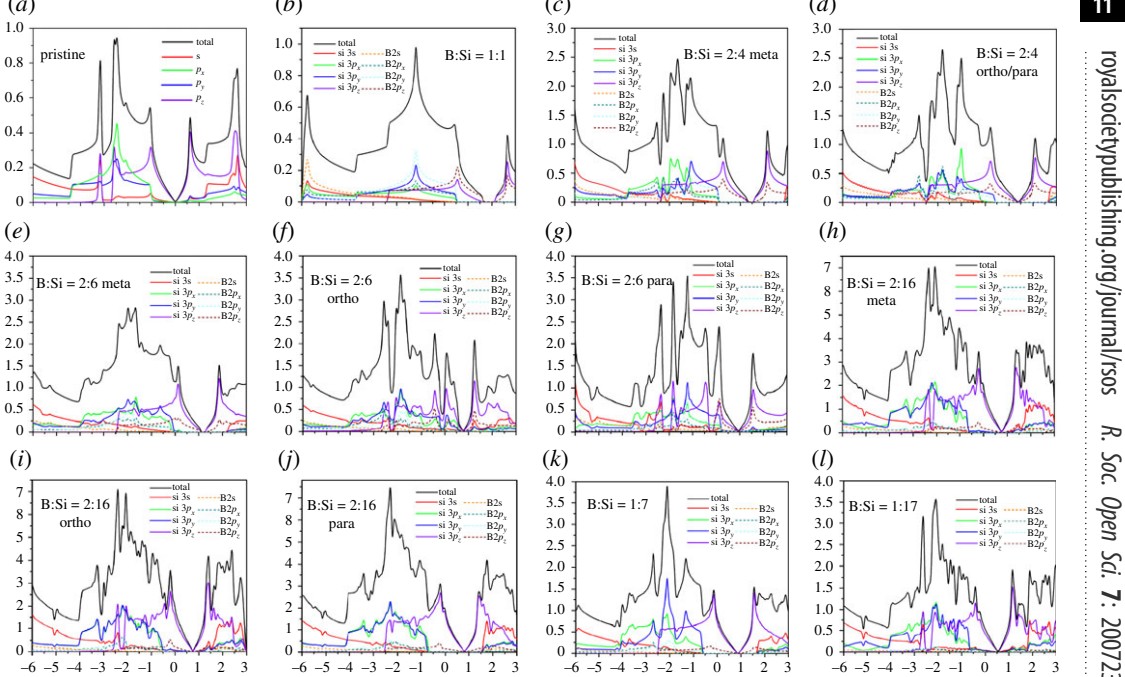

**Figure 4.** The significant van Hove singularities in [B, Si]- and orbital-projected density of states for boron-substituted silicene materials in various cases: (a) without boron-guest atoms, (b) [1:1]-, (c) [2:4]-meta-, (d) [2:4]-ortho-/para-, (e) [2:6]-meta-, (f) [2;6]-ortho-, (g) [2:6]-para-, (h) [2:16]-meta-, (i) [2:16]-ortho-, (j) [2:16]-para-, (k) [1:7]-single- and (l) [1:17]-single-configurations, where the solid/dashed red, green, blue and purple curves, respectively, correspond to [3s, $3p_x$, $3p_y$, $3p_z$]–/[2s, $2p_x$, $2p_y$, $2p_z$]-orbitals.

the final $\pi$-band shoulder structures, such van Hove singularities appear at $E \sim -2.5$ eV. Consequently, the total $\pi$-band width, being wider than 3 eV, grows with the increasing concentration. Those three kinds of special structures can characterize the $\pi$-electronic energy sub-bands, clearly illustrating the concise orbital hybridizations of $3p_z$–$3p_z$, $3p_z$–$2p_z$ and $2p_z$–$2p_z$. In addition to the strong $\pi$ peak, there also exists a prominent $\pi^*$ one above the conduction Dirac-cone structure at approximately 0.5–0.7 eV, as a result of the M-saddle-point states. Moreover, the two-dimensional free hole density is proportional to the total area underneath curve density of states between the valence Dirac-point structure and the Fermi level. When the magnitude of the unit cell is taken into account for the two-dimensional electronic state density, such area is deduced to be same as the boron concentration under any substitution case.

Generally speaking, the $\sigma$ orbitals of [2s, $2p_x$, $2p_y$] and [3s, $3p_x$, $3p_y$] make significant contributions to density of states at deeper and higher energies. The initial $\sigma$-band valence states, which mainly come from the four $p$-orbitals (the solid green/blue curves and the dashed cyan/dark cyan curves), can create an obvious shoulder structure very close to the symmetric $\pi$ peak in the range of $-0.7$ eV $< E <$ 0.2 eV. That is, the initial $\sigma$-electronic valence states exhibit the dramatic transformation from the unoccupied to the occupied ones as the B-concentration decreases from the [1:1] full compound (figure 2b–l). These two structures might be merged together at the lower concentration, e.g. the [2:16] cases in figure 2h–j. However, the $\sigma$ and $\pi$ bands do not have any anti-crossing behaviours and thus indicate the absence of sp³ bondings. With further lowering of state energy [$E < -0.7$ eV], plenty of van Hove singularities, which consist of shoulders, symmetric peaks and asymmetric ones, represent the emergences, saddle-point structures and anti-crossings of the $\sigma$-valence energy sub-bands, it might be very difficult to characterize the $\sigma$-sub-band widths from the high-resolution scanning tunneling spectroscopy (STS) measurements on their van Hove singularities, but not the initial $\sigma$-electronic energy. On the other hand, such experimental examinations are available in providing sufficient information about the $p$-type [free-hole] doings, three-/two-energy-dependent van Hove singularities of the $\pi/\pi^*$-electronic states (the initial shoulder, intermediate peak and final shoulder) and the first $\sigma$-electronic shoulder.

Since the boron-substituted silicene systems present the modified Dirac-cone structures and the well-separated $\pi$- and $\sigma$-electronic states, the tight-binding model [66] could be employed to simulate

the low-lying spin-degenerate energy dispersions. Generally speaking, an intrinsic Hamiltonian [67], which corresponds to a Moire superlattice, belongs to a very complicated Hermitian matrix even in the absence of spin-induced single- and many-particle interactions. Important factors, such as the total number of atoms, the various Si–Si, B–Si and B–B chemical bonds, and the multi-orbital hybridizations need to be taken into account simultaneously; that is, the orbital- and position-dependent hopping integrals [68] and on-site Coulomb potentials [69] should be included in the delicate calculations. Many parameters will be required to get a good fitting. Maybe, Harrison's rule [70], the strength of hopping integrals being inversely proportional to the square of the distance, is one of the efficient methods in solving the highly non-uniform chemical bonds under dilute substitutions. Then the generalized tight-binding models, in the presence of external electric and magnetic fields [71], could be quite powerful in thoroughly comprehending the rich quantization phenomena, such as the magneto-electric energy spectra [72], absorption properties [73], Hall quantum transports [49] and Coulomb excitations [74].

# 4. Concluding remarks

The geometry and electronic properties of boron-substituted silicene are studied using DFT calculations. The essential properties are enriched by boron configuration and concentration. All the final cases show metallic behaviour. The B–Si bond lengths could be evaluated via the average of B–B and Si–Si ones, which contributes to the well-separated characteristics of $\pi$ and $\sigma$ bondings. The high charge transfer from Si atoms to boron atoms leads to variations in the carrier densities that create the electronic properties change. Interestingly, the possession of three valence orbitals in boron makes all boron-substituted silicene cases demonstrate the $p$-type phenomena. That is to say, the boron substitutions are predicted to create very high two-dimensional free hole densities, leading to potential applications in nanoscaled electronic devices [75,76]. The predicted feature-rich band structures and the features in density of states could be examined by the ARPES and STS measurements, respectively.

Data accessibility. Our data are deposited at Dryad: https://doi.org/10.5061/dryad.zw3r2285w.

Authors' contributions. H.D.P.: collected data, writing, original draft. W.P.S.: conceptualization and writing, review and editing. T.D.H.N.: discussed the results. N.T.T.T. gave important comments and suggestions. M.F.L.: methodology, resources, funding and improved the writing. All authors gave final approval for publication.

Competing interests. We declare we have no competing interests.

Funding. This work was financially supported by the Hierarchical Green-Energy Materials (Hi-GEM) Research Center, from The Featured Areas Research Center Program within the framework of the Higher Education Sprout Project by the Ministry of Education (MOE) and the Ministry of Science and Technology (MOST 108-3017-F-006-003) in Taiwan.

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
