## [Reviewer comments · Royal Society Open Science]

Review History

RSOS-200723.R0 (Original submission)

Review form: Reviewer 1

Is the manuscript scientifically sound in its present form?

Yes

Are the interpretations and conclusions justified by the results?

Yes

Is the language acceptable?

Yes

Do you have any ethical concerns with this paper?

No

Have you any concerns about statistical analyses in this paper?

No

Recommendation?

Accept with minor revision (please list in comments)

Comments to the Author(s)

This manuscript presents a valuable systematic investigation on the stability and electronic properties of B-Si monolayer alloy phases. The calculation method is reliable and some general trends can be summarized. The work is helpful to the understanding of the bonding in this system, for examples the B-Si bond length, buckling and charge density distributions. Some structures, for example the 1:1 honeycomb structure presents interesting electronic structure which is worth of further experimental and theoretical studies. Overall, the manuscript has a decent quality that is suitable for publication in Royal Society Open Science.

One general question is that why there is no structure with higher B density (between 50% and 100%)? As boron can itself form monolayer structure, I expect many possible structures with boron density between 50% and 100%. Especially the authors aim to present a systematic investigation on this alloy system. Can the authors fill in this gap, or at least provide their reason why they ignore this region?

Review form: Reviewer 2**Is the manuscript scientifically sound in its present form?**

Yes

Are the interpretations and conclusions justified by the results?

No

Is the language acceptable?

Yes

Do you have any ethical concerns with this paper?

No

Have you any concerns about statistical analyses in this paper?

No

Recommendation?

Accept with minor revision (please list in comments)

Comments to the Author(s)

1. The authors mentioned that the symbol of charge transfer from silicon to boron is due to the higher carrier density. What is the fundamental mechanism?
2. The authors did calculations of band structures of different percentages of Boron atoms in silicene. However, the percentage of B atoms is as large as 33.3%. Is this a reasonable percentage of dopant for two-dimensional materials?

Review form: Reviewer 3**Is the manuscript scientifically sound in its present form?**

No

Are the interpretations and conclusions justified by the results?

Yes

Is the language acceptable?

Yes

Do you have any ethical concerns with this paper?

No

Have you any concerns about statistical analyses in this paper?

No

Recommendation?

Major revision is needed (please make suggestions in comments)

Comments to the Author(s)

Authors have presented a study of two-dimensional Silicene with Boron doping. Authors have used the first-principles study to understand the effect of Boron doping on electronic structure of Silicene with varied concentrations. Authors also mentioned about the highly non-uniform Moire super-lattices, atom-dominated band structures, charge density distributions and atom- & orbital-decomposed van Hove singularities. This study is important and will facilitate experimental investigation of electronic devices based on B-doped Silicene structures. Nevertheless, I have some concerns that should be addressed. As a summary of my report, I recommend its publication in Royal Society Open Science once the following points are taken care of.

1. The bandgap of pristine Silicene mentioned in Fig 2 (a) (0.01 eV) is not consistent with the value reported in Table 1 (0.001 eV). I suggested the authors correct this value accordingly and compare their results with previously reported results.
2. 100x100x1 k-points are taken which will be computationally very expensive. Authors need to comment on the same.
3. In Table 1, it is not clear that the Δ symbol represents which physical or chemical parameter? Please clarify.
4. Authors have inconsistently used the Brillouin zone and BZ throughout the manuscript. I suggest the authors introduce the abbreviation BZ in the starting and then follow it consistently.
5. Authors have mentioned about the buckling but it is not mentioned in the manuscript that how it varies with different B-concentrations in Silicene?
6. In literature, there is already a study on Silicene nano-ribbons with B/N doping, "Wang et al., J. Phys. Chem. C 2013, 117, 26, 13620-13626". Authors need to comment on how their study is different from the previous report. Also, I suggest the authors compare their results with the previous studies on this topic and should highlight their novel findings.
7. There is inconsistency in naming the 'B-atom', at few places it mentioned as a 'guest atom' but it's not clear that authors are taking about 'B atoms'. I suggest the authors clarify the confusion in the manuscript.
8. I suggest authors go through the manuscript more carefully as there are some writing issues such as spelling mistakes (e.g. Dirac coin). Also, authors have used "very interesting" throughout the manuscript more than 10 times which is not going well the flow of the manuscript.

Decision letter (RSOS-200723.R0)

Dear Professor Lin:

Title: Rich p-type-doping phenomena in boron-substituted silicene systems

Manuscript ID: RSOS-200723

Thank you for your submission to Royal Society Open Science. The chemistry content of Royal Society Open Science is published in collaboration with the Royal Society of Chemistry. I apologise it has taken longer than usual to send you this decision.

The editor assigned to your manuscript has now received comments from reviewers. We would like you to revise your paper in accordance with the referee and Subject Editor suggestions which can be found below (not including confidential reports to the Editor). Please note this decision does not guarantee eventual acceptance.

Please submit your revised paper before 04-Oct-2020. Please note that the revision deadline will expire at 00.00am on this date. If we do not hear from you within this time then it will be assumed that the paper has been withdrawn. In exceptional circumstances, extensions may be possible if agreed with the Editorial Office in advance. We do not allow multiple rounds of revision so we urge you to make every effort to fully address all of the comments at this stage. If deemed necessary by the Editors, your manuscript will be sent back to one or more of the original reviewers for assessment. If the original reviewers are not available we may invite new reviewers.

On behalf of the Subject Editor Professor Anthony Stace and the Associate Editor Dr Dattatray Late.

RSC Associate Editor:
Comments to the Author:
Major Revision

RSC Subject Editor:
Comments to the Author:

(There are no comments.)

Reviewers' Comments to Author:

Reviewer: 1

Comments to the Author(s)

This manuscript presents a valuable systematic investigation on the stability and electronic properties of B-Si monolayer alloy phases. The calculation method is reliable and some general trends can be summarized. The work is helpful to the understanding of the bonding in this system, for examples the B-Si bond length, buckling and charge density distributions. Some structures, for example the 1:1 honeycomb structure presents interesting electronic structure which is worth of further experimental and theoretical studies. Overall, the manuscript has a decent quality that is suitable for publication in Royal Society Open Science.

One general question is that why there is no structure with higher B density (between 50% and 100%)? As boron can itself form monolayer structure, I expect many possible structures with boron density between 50% and 100%. Especially the authors aim to present a systematic investigation on this alloy system. Can the authors fill in this gap, or at least provide their reason why they ignore this region?

Reviewer: 2

Comments to the Author(s)

1. The authors mentioned that the symbol of charge transfer from silicon to boron is due to the higher carrier density. What is the fundamental mechanism?
2. The authors did calculations of band structures of different percentages of Boron atoms in silicene. However, the percentage of B atoms is as large as 33.3%. Is this a reasonable percentage of dopant for two-dimensional materials?

Reviewer: 3

Comments to the Author(s)

Authors have presented a study of two-dimensional Silicene with Boron doping. Authors have used the first-principles study to understand the effect of Boron doping on electronic structure of Silicene with varied concentrations. Authors also mentioned about the highly non-uniform Moire super-lattices, atom-dominated band structures, charge density distributions and atom- & orbital-decomposed van Hove singularities. This study is important and will facilitate experimental investigation of electronic devices based on B-doped Silicene structures. Nevertheless, I have some concerns that should be addressed. As a summary of my report, I recommend its publication in Royal Society Open Science once the following points are taken care of.

1. The bandgap of pristine Silicene mentioned in Fig 2 (a) (0.01 eV) is not consistent with the value reported in Table 1 (0.001 eV). I suggested the authors correct this value accordingly and compare their results with previously reported results.
2. 100x100x1 k-points are taken which will be computationally very expensive. Authors need to comment on the same.
3. In Table 1, it is not clear that the Δ symbol represents which physical or chemical parameter? Please clarify.
4. Authors have inconsistently used the Brillouin zone and BZ throughout the manuscript. I suggest the authors introduce the abbreviation BZ in the starting and then follow it consistently.
5. Authors have mentioned about the buckling but it is not mentioned in the manuscript that how it varies with different B-concentrations in Silicene?
6. In literature, there is already a study on Silicene nano-ribbons with B/N doping, "Wang et al., J. Phys. Chem. C 2013, 117, 26, 13620-13626". Authors need to comment on how their study is

different from the previous report. Also, I suggest the authors compare their results with the previous studies on this topic and should highlight their novel findings.

7. There is inconsistency in naming the 'B-atom', at few places it mentioned as a 'guest atom' but it's not clear that authors are taking about 'B atoms'. I suggest the authors clarify the confusion in the manuscript.

8. I suggest authors go through the manuscript more carefully as there are some writing issues such as spelling mistakes (e.g. Dirac coin). Also, authors have used "very interesting" throughout the manuscript more than 10 times which is not going well the flow of the manuscript.

Author's Response to Decision Letter for (RSOS-200723.R0)

See Appendix A.

Decision letter (RSOS-200723.R1)

Dear Professor Lin:

Title: Rich p-type-doping phenomena in boron-substituted silicene systems
Manuscript ID: RSOS-200723.R1

It is a pleasure to accept your manuscript in its current form for publication in Royal Society Open Science. The chemistry content of Royal Society Open Science is published in collaboration with the Royal Society of Chemistry.

On behalf of the Subject Editor Professor Anthony Stace and the Associate Editor Dr Dattatray Late.

RSC Associate Editor
Comments to the Author:

(There are no comments.)

Reviewer(s)' Comments to Author:

Appendix A

Title: Rich p-type-doping phenomena in boron-substituted silicene systems

Author: Hai Duong Pham*, Wu-Pei Su, Thi Dieu Hien Nguyen, Ngoc Thanh Thuy Tran, Ming-Fa Lin*

Responses to the Referee's report

We appreciate the reviewers' significant comments on this manuscript. Considerable modifications have been made according to these comments as follows:

Reviewer: 1

Comments to the Author(s)

This manuscript presents a valuable systematic investigation on the stability and electronic properties of B-Si monolayer alloy phases. The calculation method is reliable and some general trends can be summarized. The work is helpful to the understanding of the bonding in this system, for examples the B-Si bond length, buckling and charge density distributions. Some structures, for example the 1:1 honeycomb structure presents interesting electronic structure which is worth of further experimental and theoretical studies. Overall, the manuscript has a decent quality that is suitable for publication in Royal Society Open Science.

One general question is that why there is no structure with higher B density (between 50% and 100%)? As boron can itself form monolayer structure, I expect many possible structures with boron density between 50% and 100%. Especially the authors aim to present a systematic investigation on this alloy system. Can the authors fill in this gap, or at least provide their reason why they ignore this region?

Reply: Thanks for your comments. Regarding the issue raised by the reviewer, we would like to explain as follows:

1. In fact, we are able to compute configurations at high concentrations as suggested by the reviewers. However, based on the results calculated in 12 cases, we can

easily predict the properties of the configurations between 50% and 100% B-substituted concentration. The red-shift Fermi level and modified Dirac cone might clearly be identified. Boron and silicon atoms, respectively, possess three and four valence electrons that will lead the p-type doping and create plenty of free hole near the Fermi level. The pi and sigma energy bands due to [2pz-3pz] and [2s, 2px, 2py - 3s, 3px, 3py] might well-separated and are able define the pi and sigma bandwidths. This feature implies that we can simulate first-principle results by the tight-binding model and when it succeeds, the varied fundamental properties could be explored fully in future studies, such as the rich and unique magnetic quantization phenomena, the quantum spin Hall effect, Coulomb excitation, and optical properties[1,2]. That is also the importance of this study.

2. In the experimental side, in recent studies on chemical absorption and substitution of Nitrogen on graphene [3], the highest guest-atom concentration that can be reached is 26% wt. Apparently, synthesizing structures at high concentrations such as 50%, 100% seems to be very challenged. However, our research is a theoretical study, we need to set many concentrations and configurations whether one of them is ideal. This also implies that in practice, we should focus our attention on configurations with low concentrations instead of high ones.

[1] Po-Hsin Shih et al, Carbon, 160, 211-218 (2020). DOI: 10.1016/j.carbon.2019.12.088.

[2] Thi-Nga Do et al, Phys. Rev. B, 100, 155403 (2019).

DOI: 10.1103/PhysRevB.100.155403.

[3] Jinyin Jin et al, Adv. Funct. Mater., 29, 1807441 (2019).

DOI: 10.1002/adfm.201807441.

Reviewer: 2

Comments to the Author(s)

1. The authors mentioned that the symbol of charge transfer from silicon to boron is due to the higher carrier density. What is the fundamental mechanism?

Reply: The charge transfer from silicon atom to boron atom can be realized by many features. The observable red area is located around the boron-guest atom. The increase

of Si-Si bond lengths after substitution. For example, the longer Si-Si bond length in substitution systems compare with the pristine case implies that charges from Silicone atom transfer to Boron one and make the binding between Si-Si becomes weaker, therefore, these bond lengths become longer. These features of charge transfer might be closely related to the strong affinity of the boron atoms.

2. The authors did calculations of band structures of different percentages of Boron atoms in silicene. However, the percentage of B atoms is as large as 33.3%. Is this a reasonable percentage of dopant for two-dimensional materials?

Reply:

In fact, the emergence of recent synthetic experiments shown that few-layer graphene systems can be produced by various methods, e.g., mechanical exfoliation and molecular vapor deposition. However, 2D silicene systems can be generated only with the molecular beam epitaxy (MBE). Moreover, in recent studies on chemical absorption and substitution of N on graphene [3], the highest guest-atom concentration that can be reached is 26% wt. Nevertheless, our research is a theoretical study, we need to set many concentrations and configurations whether one of them is ideal.

[3] Jinyin Jin et al, *Adv. Funct. Mater.*, 29, 1807441 (2019).

DOI: 10.1002/adfm.201807441.

Reviewer: 3

Comments to the Author(s)

Authors have presented a study of two-dimensional Silicene with Boron doping. Authors have used the first-principles study to understand the effect of Boron doping on electronic structure of Silicene with varied concentrations. Authors also mentioned about the highly non-uniform Moire super-lattices, atom-dominated band structures, charge density distributions and atom- & orbital-decomposed van Hove singularities. This study is important and will facilitate experimental investigation of electronic devices based on B-doped Silicene structures. Nevertheless, I have some concerns that should be addressed. As a summary of my report, I recommend its publication in Royal Society Open Science once the following points are taken care of.

1. The bandgap of pristine Silicene mentioned in Fig 2 (a) (0.01 eV) is not consistent with the value reported in Table 1 (0.001 eV). I suggested the authors correct this value accordingly and compare their results with previously reported results.

Reply: So sorry, that is our confusion. In our calculation, pristine silicene demonstrates a narrow bandgap of 0.001 eV at the Dirac point, mainly due to the weak spin-orbit coupling. This value is slightly smaller than the previous study results (0.001 eV with 0.015 eV) [4,5,6]. It is normal because the PBE standard always underestimates the band gap value. We have corrected it in the revised manuscript (Figure 2.a). In addition, we also have more discussion on the bandgaps of pristine silicene compared with previous studies (highlighted in revised manuscript).

[4] C. C. Liu, H. Jiang, and Y. Yao, *Physical Review B* 84, 195430 (2011).

[5] S. Chowdhury and D. Jana, *Rep. Prog. Phys.* 79 126501, (2016).

[6] C. C. Liu, W. Feng, and Y. Yao, *Phys. Rev. Letters*, 107, 076802 (2011).

2. 100x100x1 k-points are taken which will be computationally very expensive. Authors need to comment on the same.

Reply: In this study, the first Brillouin zone is sampled in the Monkhorst–Pack scheme along the two-dimensional periodic direction by 9x9x1 k-points for structure relaxations. For accurate calculation of electronic properties, we used 100x100x1 k-points for high concentration cases (50% and 100%) and 30x30x1 for lower concentration configurations (the rest). This added content has highlighted in the revised manuscript.

3. In Table 1, it is not clear that the Δ symbol represents which physical or chemical parameter? Please clarify.

Reply: The Δ symbol represents for the buckling. We also highlighted in the revised manuscript.

4. Authors have inconsistently used the Brillouin zone and BZ throughout the manuscript. I suggest the authors introduce the abbreviation BZ in the starting and then follow it consistently (highlighted in revised the manuscript).

Reply: Thank you for your suggestion, we have corrected the manuscript to follow your suggestion (highlighted in revised the manuscript).

5. Authors have mentioned about the buckling but it is not mentioned in the manuscript that how it varies with different B-concentrations in Silicene?

Reply: We have added the discussion about the buckling in the geometric structures section (highlighted in revised the manuscript).

6. In literature, there is already a study on Silicene nanoribbons with B/N doping, “Wang et al., *J. Phys. Chem. C* 2013, 117, 26, 13620–13626”. Authors need to comment on how

their study is different from the previous report. Also, I suggest the authors compare their results with the previous studies on this topic and should highlight their novel findings.

Reply: There are some previous studies about B-substituted silicene or silicene nanoribbon. However, they only concerned one concentration or configuration. Therefore, the sensitive concentration and configuration dependencies are not clear. Systematic investigations of these features are necessary. We added this content to the introduction section as a motivation for this study.

7. There is inconsistency in naming the 'B-atom', at few places it mentioned as a 'guest atom' but it's not clear that authors are taking about 'B atoms'. I suggest the authors clarify the confusion in the manuscript.

Reply: We had introduced the boron atom as a guest atom in the introduction section. Besides, we have modified "guest atom" by "B-guest atom" or "boron-guest atom" to avoid confusion. (highlighted in the revised manuscript).

8. I suggest authors go through the manuscript more carefully as there are some writing issues such as spelling mistakes (e.g. Dirac coin). Also, authors have used "very interesting" throughout the manuscript more than 10 times which is not going well the flow of the manuscript.

Reply: We have thoroughly read and checked the manuscript again to check some typos and misprint. Besides, we have modified or removed "very interestingly" to follow reviewer's suggestion.